# Unexpected Previously Unknown Diversity of the Genus *Microphor* Macquart (Diptera: Dolichopodidae: Microphorinae) in the West Palaearctic

**DOI:** 10.3390/insects13080700

**Published:** 2022-08-04

**Authors:** Miroslav Barták, Štěpán Kubík

**Affiliations:** Department of Zoology and Fisheries, Faculty of Agrobiology, Food and Natural Resources, Czech University of Life Sciences Prague, Kamýcká 129, 165 00 Praha-Suchdol, Czech Republic

**Keywords:** new species, neotype, descriptions, taxonomy, Bulgaria, Switzerland, Spain, Italy, Portugal, Turkey

## Abstract

**Simple Summary:**

The genus *Microphor* contains small to very small true flies (Diptera) well known for the kleptoparasitic behavior of females, often observed as “stealing” tiny prey from spider webs. Altogether, 22 species of the genus (including 5 described in this paper) are known worldwide, and our paper loosely follows a recent study from North America. It is quite surprising that we add five new species to seven already known from West Palaearctic, which is the most studied area in the world with respect to the Diptera.

**Abstract:**

*Microphor baechlii* sp. nov. (Switzerland, Turkey), *M. chvalai* sp. nov. (France), *M. nevadensis* sp. nov. (Spain), *M. pallipes* sp. nov. (Italy), and *M. turcicus* sp. nov. (Turkey) are described and illustrated. The neotype of *M. anomalus* (Meigen, 1824) is designated. Males of all known Palaearctic species of *Microphor* are keyed, genitalia are illustrated for new species and species previously inadequately illustrated, and main diagnostic characters are discussed. *Microphor*
*strobli* Chvála, 1986 is newly recorded from Bulgaria, *M. anomalus* (Meigen, 1824) is newly recorded from Turkey, and *M. holosericeus* (Meigen, 1804) is newly recorded from Turkey and Portugal.

## 1. Introduction

Previously, Microphorinae was classified within the former “Empididae” in the broad sense, either in the subfamily Empidinae or as a separate “*Microphorus* group” of genera in the Ocydromiinae or Hybotinae, since 1970 as a subfamily of the former “Empididae” [1]. Collin [2] was the first to use the family-level term “Microphorinae” without any specifications but instead using I.C.Z.N. Article 13.2.1; this taxon was validated by subsequent usage by Hennig [3] and Chvála [4]. Microphorinae has been elevated to family rank by Chvála [4], who considered it a clearly defined group of “empidid” flies forming, together with the family Dolichopodidae, a monophyletic subgroup of the superfamily Empidoidea. Presently, there is a broad agreement that Microphorinae is part of Dolichopodidae sensu lato [5], which also includes Parathalassiinae and Dolichopodidae sensu stricto.

Both extant and extinct taxa of Microphorinae are known. The extant taxa include *Microphor* Macquart, with 22 described species, including species described here (5 Nearctic, 15 Palaearctic, 1 Australasian, 1 Oriental), and *Schistostoma* Becker, with 49 described species (17 Palaearctic, 4 Afrotropical, 28 Nearctic); most of the Nearctic Microphorinae species were recently transferred from *Microphor* to *Schistostoma* [6]. Both genera were clearly separated [6,7], based primarily on characters of male and female terminalia. Extinct taxa were treated in detail [8].

Features of the subfamily Microphorinae were described in detail [6], including illustrations of the genitalia of Nearctic species, including five species of Nearctic *Microphor*. This description is also valid for West Palaearctic species and not repeated herewith. Characters of genitalia are the most important for distinguishing species. However, for the West Palaearctic fauna, only a few previously published illustrations of the male postabdomen of *Microphor* species are available. Chvála [1] illustrated the male genitalia of all species known to him, but without sufficient details (e.g., surstyli mostly not illustrated). Ulrich [9] depicted the postabdomen of *Microphor holosericeus* (Meigen, 1804), and Sinclair and Cumming [5] illustrated the male terminalia of *M. pilimanus* Strobl, 1899. The genitalia of newly described species, including previously inadequately illustrated species (e.g., *M. anomalus* (Meigen, 1824), *M. strobli* Chvála, 1986, and *M. rostellatus* Loew, (1864)), are presented here.

The purpose of compiling this paper is the fact that, in light of the unprecedented reduction in biodiversity and the possible further mass extinction of biota, we should describe species before they become extinct, study their ecological roles, distribution, and niches, and preserve vouchers at least in collections for further studies. In order to clarify the taxonomic status and to preserve the stability of using the species name, a neotype was designated for the *M. anomalus*. Figure 1 illustrates that, even in such a thoroughly investigated area as West Palaearctic, many new species still remain undescribed.

## 2. Materials and Methods

The material studied originated mostly from collections of the Czech University of Life Sciences, Prague (CULSP). The material was mostly collected by means of mass trapping methods (sweeping vegetation, yellow and white water pan traps) and stored in ethyl alcohol. Voucher specimens were selected and dried using the method described by Barták [10].

Genitalia preparations and drawings: genitalia, together with the preceding 2–3 abdominal segments, were removed from the rest of the body using small scissors and macerated in potassium hydroxide solution (approx. 10%) in small vials submerged in hot water for 1–2 h. After neutralizing with 8% acetic acid (5 min), the genitalia were dissected in glycerine and photographed. The photos were produced using a Nikon SMZ 1500 stereomicroscope equipped with a Canon EOS 700D digital camera. Images served as models for hand drawings, and details were added by directly observing the dissected genitalia. The resulting drawings were processed by the vector program^®^ Adobe Illustrator CC.

The morphological terms used here follow [6,11,12,13]. Genitalia details were named after [5,6]. All body measurements (including body and setae length) were taken from dry specimens (therefore, the actual length may differ from that of fresh or wet-preserved material) by means of an ocular micrometer mounted on Nikon SMZ 1500 binocular microscope. Male body length was measured from antennal base to the tip of genitalia and female body length from base of antennae to the tip of cerci. Thoracic setae are counted on one side of body (except those on scutellum and hypandrium).

Distributional records are based mostly on [14,15].

## 3. Results


**Taxonomy**


Order: Diptera Linnaeus, 1758

Family: Dolichopodidae Latreille, 1809

Subfamily: Microphorinae Collin, 1960

Genus: *Microphor* Macquart, 1827

*Microphor* Macquart, 1827: 139. Type species: *Microphor velutinus* Macquart, 1827 (=*Microphor holosericeus* (Meigen, 1804)), by subsequent designation (Rondani 1856).

*Microphorus* Macquart, 1834: 345. Unjustified emendation.

*Microphora* Zetterstedt, 1842: 253. Unjustified emendation or error, not *Microphora* Kröber, 1912: 245 (=Therevidae).

**Diagnosis**. Eyes without ommatrichia; scutellum with three or more pairs of setae, internal pair not the largest one; mid legs not modified; body dark setose; wing with cell dm present and emitting three branches, anal lobe well developed and right-angled; hypandrium with medial cleft (medial hypandrial prolongation absent), phallic process mostly present (but strongly shortened in several species), right ventral postgonite lobe hook-shaped.

The detailed description of the genus is not necessary because of existing descriptions in [1,6]. For these reasons, we concentrate on the detailed description of genitalia only. The latter paper distinguished two species groups within the genus, the *M. discalis* species group and *M. obscurus* species group. Interestingly, all West Palaearctic species have a short discal cell (distal section of vein M_1_ longer than preceding section), but the arrangement of all of them into the *M. obscurus* group may be inadequate because especially both *M. holosericeus* and *M. strobli* are strikingly different from all other species (see characters in point 4 of the key), possibly deserving their own separate group.

The male genitalia are rotated and lateroflexed to the right and inverted with the posterior end directed anteriorly. As such, the hypandrium appears in a dorsal position and the epandrial lamellae and cerci appear in a ventral position. In the following text, the terms “dorsal”, “ventral”, “anterior”, “posterior”, “right”, and “left” refer to the genitalia in its unrotated and unflexed position (i.e., with hypandrium positioned ventrally and directed posteriorly).

The genitalia are asymmetrical. Laterally, two epandrial lamellae shield nearly all internal structures, except sometimes slightly protruding parts (cerci, phallus and/or its basal process, right ventral postgonite lobe, rarely hypoproct lobe, and postgonite lobes). Cerci are almost identical in all species, without useful species-specific features and, for this reason, they are not illustrated in this study. The hypoproct is usually unsclerotized or moderately sclerotized and connected to the ventral part of cerci; in some species, it forms a separate sclerotized plate that is shorter than the cerci (*M. nevadensis* sp. nov.), and rarely do the hypoproct lobes exceed the cerci (*M. holosericeus* or Nearctic *M. discalis* Melander, [6], Figure 41). Both epandrial lamellae (right and left) are connected proximally by a narrow epandrial bridge. Epandrial lamellae usually possess dorsal, middle, and ventral surstyli lobes. Dorsal lobes may be short and broad (*M. pallipes* sp. nov., Figure 5a,b), medium long and narrow (*M. baechlii* sp. nov., *M. anomalus*, Figures 2a,b and 7a,b), or long with a submedian wart (unnamed *Microphor* sp., Figure 12a,c) or twice thickened basally (in *M. nevadensis*, Figure 4a,b, or in Nearctic *M. skevingtoni* Brooks and Cumming, 1922, [6], Figure 46). In some species, the middle lobe may be present and placed near the dorsal lobe (as in *M. rostellatus*, Figure 10a,b) or near the ventral lobe (in *M. pallipes*, Figure 5a,b), and is usually asymmetrically larger on the right lamella (in *M. nevadensis*, Figure 4a,b) or on the left lamella (*M. strobli*, Figure 11a,b). In some species, the middle lobes in the ventral position are very small, protruding slightly medially and not apparent in the lateral view (e.g., *M. anomalus*).

The ventral part (placed dorsally in intact specimens) is formed by the hypandrium, which looks like a boat cut across the posterior end (i.e., dorso-ventrally). This “cut” results in a cleft (best visible in ventral view) separating its right and left posterior parts; both parts are usually convex, forming (in lateral view) a dorsal lobe and a ventral lobe (and because they are positioned apically, are referred to below as dorso- and ventro- apical). Dorsoapical lobes on both the right and left side are usually (except, e.g., in *M. holosericeus*) more or less tightly connected to the dorsal postgonite lobes; ventroapical lobes are differently shaped on the left and right side: the left side is usually smoothly curved and shorter, and the right side is usually longer, protrudes mostly into a more or less elongate curved and tipped process, or is rounded (e.g., *M. strobli*, Figure 11e). The anterior part of the hypandrium (from base to median split) bears a different number of long and strong setae usually arranged in two rows. The number of these setae ranges from four (two setae on each side) to eighteen (*M. zimini* Shamshev, 1995) and, in most species, their number is variable, often different on the left and right side, and sometimes even placed along the midline (therefore, they are not illustrated here, except their insertion points). Rarely (*M. turcicus* sp. nov.) does the anterior part of the hypandrium bear a transverse row of setae. Interestingly, the arrangement of these setae is less diverse in Nearctic species, where, altogether, 6–12 setae occur in two sometimes irregular rows (i.e., 3–6 setae in each row). Hypandrial lobes sometimes bear short or longer setae (e.g., unnamed *Microphor* sp. Has a rather long spine-like seta on the dorsoapical lobe, Figure 12b,e).

If the epandrial lamellae are removed, the internal components are visible and can be separated from the hypandrium. The internal components are composed of two main structures: the phallus and postgonites. The phallus has a distinct ejaculatory apodeme at its base and a basal process opposite of the apodeme. This basal process is, in most species, rod-like and narrowed apically, nearly straight (curved only at the apex), or deeply curved (*M. crassipes*, Figure 8f); in *M. holosericeus* or Nearctic *M. skevingtoni* Brooks and Cumming, 2022, [6], (Figure 47), it is short and nearly square-like, and in *M. strobli* it is apparently absent (Figure 11f). Postgonites bear paired dorsal and ventral lobes and apodemes. Dorsal lobes are usually asymmetrical, with the right lobe stronger than the left lobe, which may be reduced to a slender scarcely sclerotized rod-like structure (*M. chvalai* sp. nov., Figure 3c), and are rarely similar to each other (*M. nevadensis* sp. nov., Figure 4c). Both right and left dorsal postgonite lobes are attached to the dorsoapical hypandrial lobes; sometimes, they possess another structure at the tip, which is free and ball-like (in *M. holosericeus*) or tightly attached to the dorsoapical hypandrial lobes (in *M. chvalai* or *M. nevadensis*). After preparation, these structures may remain attached to the dorsoapical lobes of the hypandrium (as in Chvála [1], Figure 35) or to the dorsal postgonite lobes (as in Figure 4c, *M. nevadensis*). Ventral postgonite lobes are composed of right and left lobes, of which, mostly only the right lobe is fully developed and S-shaped or T-shaped (and rarely almost triangular, as in *M. rostellatus*, Figure 10c, or “A”-shaped or narrow, similar to the phallus, e.g., *M. skevingtoni* Brooks and Cumming 2022, Figure 45). The left ventral postgonite lobe is absent in some species (*M. anomalus* and allies), developed as a thin rod (*M. holosericeus*) similar to the right lobe but smaller (e.g., in *M. strobli*, Figure 11c). Paired and usually symmetrical postgonite apodemes ([6], Figures 73 and 74) are developed in all species, and are rod-like or somewhat widened apically to nearly triangular (e.g., in *M. chvalai*, Figure 3c).


**Descriptions of new species**



**
*Microphor baechlii*
**
**sp. nov.**



Figure 2


**Type material: HOLOTYPE** ♂, Turkey: Akyaka, forest, 30 m, SW [=sweeping vegetation], 37°03′16″ N, 28°19′35″ E, Barták, Kubík, 30.iv.-9.v.2013. **Paratypes**: 3 ♂, Turkey: Akyaka, pasture, 6 m, 37°03′19″ N, 28°20′07″ E, Barták, Kubík, 28.4.-8.5.2013; 1 ♂, Switzerland: VS, Jeizinen [=46°20′ N, 7°44′ E, 1740 m], 13.viii.2013, G. Bächli leg.—(CULSP).

**Other material** (excluded from type series): Turkey: 1 ♀, Akyaka, pasture, 6 m, 37°03′19″ N, 28°20′07″ E, Barták, Kubík, 28.4.-8.5.2013; 3 ♀, Akyaka, salty meadow, SW + PT [=sweeping vegetation + water pan traps], 37°02′53″ N, 28°19′39″ E, 28.iv.-9.v.2013—(CULSP).

**Diagnosis:** Black species of *Microphor* with narrow hind tibia and basitarsus and long proboscis. Acrostichals almost regularly quadriserial and dorsocentrals irregularly biserial. Mesoscutum velvety black in dorsal view. Mid femur with short posteroventral setae on basal part and contrastingly long on apical part.

**Etymology:** The species epithet is a patronym honouring Swiss dipterist Gerhard Bächli, collector of one paratype.

**Description: Male head** black, holoptic, entirely dark brownish grey microtrichose, black setose. Eye large, dorsal and ventral parts divided by blackish line, larger dorsal part with much larger facets than smaller ventral part. Occiput sparsely covered with setae up to 0.15 mm long dorsally and shorter ventrally, several longer setae alongside oral margin. Frons (very small triangle above antennae) without setae. Face including clypeus and gena microtrichose, without setae, face small and clypeus longer. Both anterior and posterior pairs of ocellar setae short (approximately 0.10 mm), with another very small pair between them. Palpus brown, elongated (0.25 mm long) and narrow, with several rather long setae; labrum long (approximately 0.40 mm, slightly shorter than head height), postmentum subshiny, elongated, with several setae; labellum with much shorter setae than palpus. Antenna black, scape without setae, pedicel with circlet of short setae (0.08 mm); postpedicel without distinct basal thickening; length of antennal segments (scape: pedicel: postpedicel: stylus, in 0.01 mm scale) = 5: 5–6: 22–25: 10–15. **Thorax** black, mesoscutum in anterior view rather light grey on anterior part, with conspicuous velvety black stripes below dorsocentrals and black (somewhat less strikingly velvety) stripe below acrostichals; in dorsal and posterior view, whole mesoscutum velvety black with narrow lighter lines between acrostichals and dorsocentrals; pleura distinctly lighter than mesoscutum. Chaetotaxy: antepronotum with one small seta on each side, proepisternum with one to –two short setae, propleura with long seta; prosternum isolated from proepisternum, without setae; postpronotum with one long and strong seta and several much shorter setae; four setae (subequally long as seta on propleura) on anterior part of mesoscutum directed obliquely upwards; acrostichals almost regularly four serial and short (length of setae in middle of rows approximately 0.05 mm), eight to eleven setae in each row; dorsocentrals irregularly biserial (last seta long and prescutellar pair very long); one presutural intraalar, one presutural supraalar; notopleuron with two long setae and a row of three to four rather long interstitials; one supraalar, one prealar, and one postalar seta; three pairs of scutellars (subapical seta long, shorter setae relatively long). Laterotergite bare. **Legs** including coxae blackish brown, microtrichose, black setose. Fore femur dorsally short setose, anteroventral setae very short on basal part and practically absent on apical half, posteroventral setae elongated only on apical third. Fore tibia narrow, without longer setae. Mid femur similarly setose as fore femur, several rather strong posteroventrals on apical third 1.5×X longer than femur depth, much shorter on basal part. Mid tibia narrow, short setose. Hind femur dorsal with setae slightly shorter than femur depth, anteroventral setae much shorter than femur depth on basal two thirds, three to four on apical third slightly longer than femur depth. Posteroventrals very short, with two to three somewhat longer on apical third. Hind tibia equally narrow, all setae shorter than tibia depth. All basitarsi slender, without conspicuous setae. **Wing** membrane clear or only indistinctly smoky, covered with microtrichia, veins brown, Sc complete, merging apically with sclerotization between Sc and R_1_ but visible up to its end. Stigma brown, at widest point occupying 3/4 of cell R_1_, anal vein visible only at extreme base, axillary angle obtuse, both basal costal setae (dorsal and ventral) black, long and strong. Halter brown with black stem, calypter creamy with long brown fringes. **Abdomen** brownish black (almost velvety brownish black in dorsal view), microtrichose, rather short black setose. Lateral marginal setae on tergites shorter than segments (approximately 0.15 mm long); discal setae slightly shorter. Sternite 1 without setae, sternites 2–5 very short setose, sternite 5 with a single rather longer lateral hind marginal seta. Genitalia (Figure 2) similar in general shape to *M. anomalus*. Phallus C-shaped with slightly bent and apically narrowed basal process (Figure 2f); both epandrial lamellae with narrow finger-like dorsal processes of surstyli (Figure 2a,b), left lamella with conspicuous ventral process; hypandrium with four (two pairs) setae, right ventroapical lobe short and simply bowed (Figure 2e, longer and irregularly curved in *M. anomalus*). **Length:** body 1.9–2.1 mm, wing 1.9–2.2 mm.

**Female:** similar to male. Head dichoptic, somewhat lighter grey than in male, ventral part of eye (below line) larger than dorsal part, all facets subequal in size. Frons approximately 0.15 mm wide at middle. Mesoscutum rather light grey microtrichose in anterior view; in dorsal view, with distinct dark stripes below acrostichals and dorsocentrals (ending anteriorly from the first prescutellar pair). Legs without conspicuous setae. Halter brown. Abdomen shorter setose than in male.

**Remarks:** The species described above is similar to other species of the *M. rostellatus* complex (see discussion below on *M. nevadensis* sp. nov.). Males of this complex may be distinguished according to the key. In the key by Chvála [1], females lead to *M. rostellatus* (due to its proboscis being only slightly shorter than its head height); however, the newly described species has distinct dark spots on the mesoscutum in the dorsal view and darker halteres (in *M. rostellatus*, the mesoscutum has no distinct pattern and the halters are brownish yellow).


***Microphor chvalai* sp. nov.**


Figure 3.

**Type material: HOLOTYPE** ♂, France: 5 km N of Sété, macchia + ruderal, PT [=water pan traps], 43°27′1″ N, 3°42′53″ E, Barták, 21-23.v.2006. **Paratypes**: 3♂, 9 km N of Sété, seashore dunes, PT, 43°21′9″ N, 3°45′48″ E, Barták, 21–23.v.2006—(CULSP).

**Diagnosis:** Black species of *Microphor* with swollen hind tibia and basitarsus. Labrum long (as long as eye height), halter yellow. Fore femur with very long posteroventrals. Acrostichals quadriserial and very short, dorsocentrals irregularly biserial and distinctly longer.

**Etymology:** The species epithet is a Latin genitive patronym in honour of Prof. Milan Chvála (who unfortunately died during the preparation of this manuscript) that recognises his contribution to the knowledge of Diptera, including the genus *Microphor*.

**Description: Male head** brownish black, holoptic, entirely dark grey microtrichose, black setose. Eye large, dorsal and ventral parts divided by blackish line, dorsal part somewhat larger with distinctly larger facets than ventral part. Occiput sparsely covered with medium long setae (approximately 0.10 mm dorsally), sparsely covered with short setae on mid and lower part, several longer setae on sides of oral cavity. Frons (very small triangle above antennae) without setae. Face microtrichose, without setae, face small (approximately 0.06 mm long and 0.10 mm wide), clypeus somewhat longer (0.09 mm long and 0.07 mm wide), both clypeus and gena subshiny. Anterior pair of ocellar setae approximately 0.10 mm long, posterior two pairs much shorter. Palpus brown, narrow, with several rather long setae, 0.22 mm long; labrum long (approximately 0.45 mm, equally long as eye height), postmentum microtrichose, very long (as long as labrum, 10×x longer than wide) with two pairs of setae; labellum covered with shorter setae than palpus. Antenna black, scape without setae, pedicel with circlet of short setae (0.05 mm); postpedicel not swollen basally; length of antennal segments (scape: pedicel: postpedicel: stylus, in 0.01 mm scale) = 4–5: 5–6: 20–22: 14–15. **Thorax** black, rather light grey microtrichose, mesoscutum with brownish tinge; in anterior view, with indication of darker stripe below acrostichals; in posterior view, slightly lighter in front of scutellum and with indications of two lighter stripes between rows of setae. Chaetotaxy: antepronotum with one very small seta on each side, propleuron with one seta, proepisternum with one very short seta; prosternum isolated from proepisternum, without setae; postpronotum with one long and strong seta and several much shorter setae; four small setae on anterior part of mesoscutum just behind antepronotum; acrostichals quadriserial, short and fine (length of setae in middle of rows approximately 0.04 mm), eight to ten setae in each row; dorsocentrals irregularly biserial and distinctly longer than acrostichals (0.10 mm in middle of rows, prescutellar pair very long); one long presutural intraalar, one long presutural supraalar; notopleuron with two setae and two to three much smaller setae between them; one supraalar, one prealar (and several additional setae in prealar area), and one postalar seta; three pairs of scutellars (middle one—subapical—very long, remaining much shorter). Laterotergite bare. **Legs** including coxae blackish brown microtrichose, black setose. Fore femur dorsally short setose, anteroventral row of setae on proximal part slightly shorter than femur depth and very short on distal half, posteroventral row of very long setae (up to 0.20 mm long in middle or row, nearly twice longer than femur depth). Fore tibia narrow, posterodorsal setae slightly longer than tibia depth. Mid femur short setose dorsally, anteroventral setae short on basal part and somewhat longer (as long as femur depth) on apical third, posteroventrals on apical half subequally long as femur depth. Mid tibia slightly widened, with several anterodorsal setae slightly longer than tibia depth, ventral setae short, on apical half distinctly spinose. Hind femur dorsally on proximal half with setae slightly shorter than femur depth, anteroventral setae short on basal part and somewhat longer (approximately as long as dorsal ones) on apical part, posteroventrals remarkably short. Hind tibia distinctly swollen apically, slightly broader than hind femur, dorsally with a row of setae as long as tibia depth. Basitarsi of fore and mid legs slender, hind basitarsus slightly but distinctly swollen, remaining joints of all tarsi slender with short setae. **Wing** membrane clear, entirely covered with microtrichia, veins yellowish brown, Sc incomplete, merging apically with sclerotization between Sc and R_1_. Stigma brown, at widest point occupying slightly more than half of cell R_1_, anal vein almost absent, axillary angle obtuse, both basal costal setae long and strong. Halter yellow, calypter dirty whitish with brown fringes. **Abdomen** brownish black, microtrichose, black setose. Lateral marginal setae on tergites approximately as long as segments (approximately 0.18 mm long); discal setae shorter. Two pairs of posterior marginal setae on sternites (especially 4–5) subequally long as those on tergites, lateral marginal and discal setae shorter. Sternite 1 without setae. Genitalia as in Figure 3. Both epandrial lamellae with very narrow dorsal processes of surstyli (Figure 3a,b), left lamella with small additional dorsal process; phallus slightly swollen and irregularly curved apically, with short, narrow, and slightly curved basal process (Figure 3f); right ventral postgonite lobe S-shaped, dorsal postgonite lobes tightly attached to hypandrial processes (Figure 3c, on figure: the connecting structure remained attached to dorsoapical lobe of hypandrium on left side and on dorsal postgonite process on right side), and triangular postgonite apodemes; hypandrium with three pairs of setae on each side, right ventroapical lobe irregularly curved (Figure 3d). **Length:** body 1.9–2.3 mm, wing 1.8–2.0 mm.

**Female:** Unknown.

**Remarks:** The species described above has some similarities with *M. gissaricus* Shamshev, 1992, [16] (swollen hind tibiae and yellow halter); however, it may be differentiated by characters given in the key.


**
*Microphor nevadensis*
**
**sp. nov.**


Figure 4.

**Type material: HOLOTYPE** ♂, Spain: Sierra Nevada, 1400 m, Trevelez, nr. river, SW [=sweeping vegetation], 37°0′9″ N, 3°15′43″ W, Barták, 15.viii.2006. **Paratypes**: 2 ♂, Spain: Sierra Nevada Mts, SW, Mecina Bombaron, nr. river, 1100 m, 36°59′20″ N, 3°9′0″ W, Barták, 18.viii.2006—(CULSP).

**Other material** (excluded from type series): 1 ♀ same data as holotype, 1 ♀ same data as paratype (CULSP).

**Diagnosis:** Black species of *Microphor* with narrow hind tibia and basitarsus and long labrum. Acrostichals irregularly three to four serial and dorsocentrals irregularly biserial. Both mid and hind femur with conspicuously long and strong posteroventral setae. Postpedicel without conspicuous basal thickening. Halter brown with black stem.

**Etymology:** The species epithet is a toponym derived from type locality.

**Description: Male head** black, holoptic, entirely dark grey microtrichose, black setose. Eye large, dorsal and ventral parts divided by blackish line, dorsal part somewhat larger with much larger facets than ventral part. Occiput sparsely covered with setae approximately 0.10 mm long dorsally and shorter ventrally. Frons (very small triangle above antennae) without setae. Face including clypeus and gena microtrichose, without setae, face small and clypeus longer. Anterior pair of ocellar setae medium long (approximately 0.15 mm), posterior pair shorter, another very small pair between them. Palpus brown, elongated (up to 0.30 mm long) and narrow, with several rather long setae; labrum long (approximately 0.40 mm, slightly shorter than head height), postmentum subshiny, elongated, with three to four pairs of rather long setae; labellum with much smaller setae than palpus. Antenna black, scape without setae, pedicel with circlet of short setae (0.06 mm); postpedicel without distinct basal thickening; length of antennal segments (scape: pedicel: postpedicel: stylus, in 0.01 mm scale) = 5–6: 5–6: 18–21: 13–15. **Thorax** black, dark brownish grey microtrichose, mesoscutum dark black and very slightly subshiny, without stripes (except usual lighter colour of prescutellar depression). Chaetotaxy: antepronotum with one small seta on each side, both proepisternum and propleuron with short seta; prosternum isolated from proepisternum, without setae; postpronotum with one long and strong seta and several much shorter setae; four small setae on anterior part of mesoscutum directed obliquely upwards; acrostichals irregularly three to four serial (length of setae in middle of rows approximately 0.08 mm), eight to nine setae in each row; dorsocentrals irregularly biserial, approximately nine to ten setae in each row, including last two longer pairs—prescutellar pair very long); one presutural intraalar, one long presutural supraalar; notopleuron with two setae (anterior one usually much smaller) and one to three additional smaller setae between them; one supraalar, prealar indistinct, and one postalar seta; three pairs of scutellars (only one pair usually long, remaining shorter). Laterotergite bare. **Legs** including coxae blackish brown, microtrichose, black setose. Fore femur dorsally short setose, anteroventral row of setae on proximal part slightly shorter than femur depth and very short on distal half, posteroventral row of setae slightly longer than femur depth. Fore tibia narrow, without longer setae. Mid femur short setose dorsally, three to four somewhat longer anteroventral setae on apical part, a row of seven to ten strong posteroventrals 1.5×X longer than femur depth (much shorter on proximal third to fourth). Mid tibia narrow, short setose. Hind femur dorsally with setae as long as femur depth, anteroventral setae much shorter than femur depth and a complete row of posteroventral setae slightly longer than femur depth. Hind tibia narrow (only very slightly widened towards apex but still narrower at broadest point than hind femur), dorsal setae shorter than tibia depth. All basitarsi slender, without conspicuous setae. **Wing** membrane clear or only indistinctly smoky, covered with microtrichia, veins blackish brown, Sc nearly complete, merging apically with sclerotization between Sc and R_1_. Stigma dark brown, at widest point occupying 3/4 of cell R_1_, anal vein visible only on extreme base, axillary angle obtuse; both basal costal setae (dorsal and ventral) black, long and strong. Halter brown with black stem, calypter dirty whitish with long brown fringes. **Abdomen** brown (almost velvety blackish brown from some points of view), microtrichose, long black setose. Lateral marginal setae on tergites longer than segments (up to 0.30 mm long); discal setae shorter. Sternite 1 without setae, sternites 2–3 short setose, sternites 4–5 each with one to –two strong and long curved lateral seta(e) and additional shorter setae. Genitalia (Figure 4) remarkably similar to *M. pilimanus.* Dorsal lobes of both (right and left one) surstyli twice thickened on basal half (Figure 4a,b); the right middle lobe of surstyli protrudes dorsally and extends beyond the epandrium as thin and pointed rod, the left one very short; hypoproct slightly shorter than cerci; phallus simply C-shaped, with rather long basal process (Figure 4f); dorsal lobes of postgonites at apex T-shaped, firmly attached to dorso-apical hypandrial lobes on both sides (Figure 4c); hypandrium with two to three pairs of setae, right ventroapical lobe short and simply curved (Figure 4e). **Length:** body 2.0–2.1 mm, wing 2.0–2.2 mm.

**Female:** similar to male. Head dichoptic, ventral part of eye (below line) larger than dorsal part. Frons approximately 0.15 mm wide at middle, slightly subshiny. Mesoscutum black, microtrichose, without stripes. Legs without conspicuous setae. Halter yellow. Abdomen much shorter setose than in male.

**Remarks:** The species described above is similar to other species of *M. rostellatus* complex. Males share several characters: narrow hind tibia, long proboscis, dark halter, three to four serial acrostichals and irregularly biserial dorsocentrals. Genitalia are similar to those of *M. pilimanus*; however, the newly described species has a narrow hind leg with slender basitarsus as long as three following segments (two segments in *M. pilimanus*), much longer labrum, and fore basitarsus lacking specialised setae (dense brush of ventral setae not developed). Females have few distinct features, so they were associated with males only according to the same sampling events. This is also why they were not included in the type series. In the key by Chvála [1], females lead to *M. rostellatus* (due to the proboscis being only slightly shorter than the head height); however, this species differs from *M. nevadensis* sp. nov. by its light grey body.


**
*Microphor pallipes*
**
**sp. nov.**


Figure 5.

**Type material: HOLOTYPE** ♂, Italy: Terni—7 km SEE, Lago di Piediluco, SW [=sweeping vegetation], damp meadow, 42°33′ N, 12°45′ E, Barták, 21.viii.2004. **Paratypes**: 12 ♂, 5 ♀ same data as holotype—(CULSP).

**Diagnosis:** Very small brownish black species of *Microphor* with swollen hind tibia and basitarsus. Antennal style slightly longer than postpedicel. Legs pale, brownish yellow, and yellow in female. Acrostichals biserial and dorsocentrals uniserial.

**Etymology:** The species epithet is derived from very pale legs.

**Description: Male head** brownish black, holoptic, entirely dark grey microtrichose, black setose. Eye large, dorsal and ventral parts divided by blackish line, dorsal part somewhat larger with much larger facets than ventral part. Occiput sparsely covered with very short setae (approximately 0.05 mm), several longer setae on sides of oral cavity. Frons (very small triangle above antennae) without setae. Face including clypeus and gena microtrichose, without setae, face small (approximately 0.05 mm long and 0.10 mm wide), clypeus larger. Anterior pair of ocellar setae short (approximately 0.07 mm), posterior pair much shorter. Palpus brown, narrow, with several rather long setae; labrum short (approximately 0.13 mm, subequally long as palpus), postmentum microtrichose very short (0.05 mm), with several setae; labellum covered with smaller setae than palpus. Antenna brown, scape without setae, pedicel with circlet of short setae (0.05 mm); postpedicel distinctly swollen basally and then suddenly narrowed; length of antennal segments (scape: pedicel: postpedicel: stylus, in 0.01 mm scale) = 3–4: 5–6: 19–20: 21–24. **Thorax** blackish brown, dark brownish grey microtrichose; in posterior view, slightly lighter in front of scutellum and with indications of two lighter stripes between rows of setae. Chaetotaxy: antepronotum with one very small seta on each side, proepisternum with short seta; prosternum isolated from proepisternum, without setae; postpronotum with one long and strong seta and several much shorter setae; two very small setae on anterior part of mesoscutum; acrostichals biserial, short and fine (length of setae in middle of rows approximately 0.05 mm), only four to six setae in each row; dorsocentrals uniserial, much longer and stronger than acrostichals (0.12 mm in middle of rows but longer posteriorly, only four to six setae in each row); one presutural intraalar, one long presutural supraalar; notopleuron with two to three setae, one to two of them usually short; one supraalar, one prealar, and one postalar seta; three pairs of scutellars (only one pair usually long, remaining very short). Laterotergite bare. **Legs** including coxae yellowish brown to brownish yellow, microtrichose, black setose. Fore femur dorsally short setose, anteroventral row of setae on proximal part slightly shorter than femur depth and very short on distal half, posteroventral row of setae nearly as long as femur depth or even longer distally. Fore tibia narrow, without longer setae. Mid femur short setose dorsally, two to three somewhat longer anteroventral setae on basal part, posteroventrals on apical half strong, subequally long as femur depth. Mid tibia narrow, short setose, setae in anteroventral row spinose. Hind femur dorsally on proximal half and anteroventrally on distal half with setae as long as femur depth. Hind tibia distinctly swollen apically, distinctly broader than hind femur, anterodorsal complete row of upright standing long setae as long or (on proximal half) longer than tibia depth. Basitarsi of fore and mid legs slender, mid one with rather spinose ventral setae, hind basitarsus slightly but distinctly swollen. **Wing** membrane clear, covered with microtrichia, veins brownish yellow, Sc nearly complete, merging apically with sclerotization between Sc and R_1_. Stigma only slightly darkened, at widest point occupying 2/3 of cell R_1_, anal vein almost absent, axillary angle obtuse; basal costal seta (dorsal) long and strong, ventral usually much smaller. Halter brown, calypter dirty whitish with long brown fringes. **Abdomen** brown (almost velvety blackish brown from some points of view), microtrichose, sparsely black setose. Lateral marginal setae on tergites approximately as long as segments; discal setae shorter. Posterior marginal setae on sternites (especially 3–5) nearly as long as those on tergites. Sternite 1 without setae. Genitalia as in Figure 5. Phallus slightly curved, without basal process (Figure 5c); both epandrial lamellae with dorsal processes of surstyli short and broad (Figure 5a,b), middle lobe of surstylus on left side long with narrow tip protruding ventrally; right ventral postgonite lobe S-shaped (Figure 5e); hypandrium usually with six setae (three pairs, ranging from five to seven), right ventroapical lobe elongated, finger-like, and apically rounded (Figure 5d). **Length:** body 1.7–2.0 mm, wing 1.8–2.0 mm.

**Female:** similar to male. Head dichoptic, brownish black, light grey microtrichose. Eye smaller than in male, size differences between lower and upper facets are not so striking. Frons approximately 0.15 mm wide at middle. Ocellars longer than in male, ocellar triangle with two pairs of additional very small setae. Mesoscutum brown, grey microtrichose, with less distinct stripes in posterior view. Legs yellow, without conspicuous setae. Halter yellow. Abdomen with much shorter setose than in male.

**Remarks:** The species described above is, in several characteristics, similar to *M. strobli*: a long stylus, short labrum, biserial acrostichals, uniserial dorsocentrals, basally swollen postpedicel, small size, general shape of the epandrial lamellae and phallus, and absent or very short phallic process; however, the male hind tibia is distinctly strongly swollen and strongly setose dorsally (similar to species of the *M. anomalus* complex), but species of this complex have multiserial mesoscutal setae, except possibly *M. sinensis* Saigusa and Yang, 2002, but this species (insufficiently illustrated) has black legs and triserial acrostichals. There are several other characteristics distinguishing *M. strobli* and *M. pallipes*; for example, the acrostichals are smaller than the distance between acrostichals and dorsocentrals in *M. pallipes* but longer in *M. strobli*; the fore femur has a row of elongated posterodorsals in *M. strobli* that is absent in *M. pallipes;* and the mid femur has uniform posteroventral setae in *M. strobli* but they are strikingly stronger and longer on the distal third in *M. pallipes*. There are also many differences in male genitalia between the two species. Females lead to *M. strobli* in the key by Chvála [1], but differ in their yellow legs and halteres.


**
*Microphor turcicus*
**
**sp. nov.**


Figure 6.

**Type material: HOLOTYPE** ♂, Turkey: Toparlar, lowland forest, SW + PT [=sweeping vegetation plus water pan traps], 8 m, 36°59′27″ N, 28°38′50″ E, Barták, Kubík, 28-30.iv.2016. **Paratypes**: Turkey: 2 ♂, same data as holotype; 1 ♂ Akyaka, pasture, 6 m, 37°03′19″ N, 28°20′07″ E, 28.iv.-8.v. 2013, Barták, Kubík; 1 ♂ Akyaka, pasture, 8 m, 37°03′11″ N, 28°20′33″ E, 27.iv.2016, Barták, Kubík—(CULSP).

**Diagnosis:** Black species of *Microphor* with slender hind tibia. Fore tibia with dorsal preapical seta longer than basitarsus. Style very short, approximately ¼ as long as postpedicel. Sternite 7 and ventral parts of tergite 6 lustrous. Both acrostichals and dorsocentrals multiserial, not distinctly separated by bare stripe. Basal part of hypandrium with approximately eight setae arranged in an irregular transverse row.

**Etymology:** The species is named after the country of origin (Turkey).

**Description: Male head** black, holoptic, entirely dark grey microtrichose, black setose. Eye large, dorsal and ventral parts (almost equally large) divided by blackish line, dorsal third with larger facets than ventral part. Occiput with postocular row nearly complete, setae in dorsal part somewhat longer than ocellars; behind this row, only sparse shorter setae, except several long setae just behind oral cavity. Frons (very small triangle above antennae) without setae. Face including clypeus and gena microtrichose, without setae, face small (approximately 0.10 mm long and 0.13 mm wide), clypeus subequally long but slightly narrower, with broad membranous parts anteriorly and on sides. Anterior pair of ocellar setae short (approximately 0.10 mm), posterior pair much shorter; on posterior part of ocellar triangle, another pair of setae longer than anterior pair. Palpus brown, narrow, with several ordinary setae; labrum short (approximately 0.25 mm, approximately half as long as head height and slightly longer than palpus), postmentum microtrichose, labellum long and broad, covered with similar setae as palpus. Antenna black, scape without setae, pedicel with circlet of short setae (0.06 mm); postpedicel narrowly triangle-shaped; length of antennal segments (scape: pedicel: postpedicel: stylus, in 0.01 mm scale) = 5–6: 7–8: 26–27: 7–8. **Thorax** black, grey microtrichose, mesoscutum dark black and slightly subshiny from all points of view. Chaetotaxy: antepronotum with one rather long seta on each side, both propleuron and proepisternum with long and strong seta; prosternum isolated from proepisternum, without setae; postpronotum with one long and strong seta and several additional much shorter setae; four strong spinose setae on anterior part of mesoscutum standing perpendicular to mesoscutum, directed towards occiput; acrostichals irregularly quadriserial, only indistinctly separated from multiserial dorsocentrals, leaving only narrow lateral parts and long prescutellar depression without setae (length of setae in middle of rows approximately 0.09 mm), prescutellar pair very long, even preceding one to two dorsocentrals longer than remaining; zero to one presutural intraalar, one long and strong presutural supraalar; notopleuron with two strong and long setae and additional two to three small setae between them; one supraalar, one small prealar (indistinctly differentiated from nearby setae), one postalar; three pairs of scutellars, mid one (subapical) very long. Laterotergite bare. **Legs** including coxae black, microtrichose, black setose. Fore femur slightly spindle-shaped dilated, dorsally short setose, anteroventral row of setae on proximal part slightly shorter than femur depth and very short on distal half, posteroventral row of setae nearly as long as femur depth. Fore tibia narrow, with short setae, preapical dorsal seta conspicuously long, slightly longer than fore basitarsus. Mid femur short setose dorsally and anteroventrally, somewhat longer anterior seta on basal third, posteroventrals slightly shorter than femur depth. Mid tibia narrow, short setulose, with single rather long subbasal anterodorsal seta. Hind femur dorsally with setae less than half as long as femur depth, anteroventrals shorter in middle and somewhat longer on distal third. Hind tibia narrow, not at least swollen, short setose (all setae shorter than tibia depth), except one subbasal and one subapical anterodorsal setae, both slightly longer than tibia depth. Basitarsi of all legs slender, without conspicuous setae. **Wing** very slightly clouded, covered with microtrichia, Sc nearly complete, merging apically with sclerotization between Sc and R_1_. Stigma brown, at widest point occupying slightly more than half of cell R_1_, anal vein incomplete, axillary angle slightly obtuse, basal costal setae (both dorsal and ventral) long and strong. Halter brown, calypter dirty whitish with long brown fringes. **Abdomen** black, microtrichose, only very slightly subshiny, black setose. Lateral (ventral) parts of tergite 6 and sternite 7 shiny. Lateral marginal setae on tergites approximately as long as segments; discal setae shorter. Lateral marginal setae on sternites (especially 3–5) nearly as long as those on tergites. Sternite 1 without setae. Genitalia as in Figure 6. Dorsal lobes of both surstyli (left and right) short and wide (Figure 6a,c); right ventral process of surstyli with very long seta (0.15 mm long); phallus irregularly slightly curved and narrow at apex (Figure 6g); hypandrium on basal part with transverse row of approximately seven strong setae, more apically with several much smaller setae and with another group of small submedian setae, left dorsoapical lobe with an additional seta (Figure 6b); postgonite of very unusual shape. **Length:** body 2.3–2.6 mm, wing 2.1–2.3 mm.

**Female:** Unknown

**Remarks:** The new species described above may be easily recognised according to several unique characters given in the key and diagnosis above. The genitalia possess several very peculiar characteristics, unknown in the remaining species: a very long seta on the ventral process of the right epandrial lamella, a transverse row of eight strong setae on the basal part of the hypandrium, etc.


**Key to males of West Palaearctic species of *Microphor* (including two species from Central Asia).**
1.Fore basitarsus with a brush of long setae ventrally. (Additional characters: acrostichals irregularly three to four serial, dorsocentrals one to two serial; hypandrium with two pairs of setae, see [1], Figure 35…………………………….. ***pilimanus* Strobl**
–Fore basitarsus without brush of long setae ventrally.……………………………22.(1) Hind tibia not distinctly swollen, being narrower than hind femur.………….……3
–Hind tibia distinctly swollen, being at least as wide near the tip as hind femur……………………………………………………………………………………..93.(2) Fore tibia with dorsal preapical seta longer than basitarsus. Style very short, approximately ¼ as long as postpedicel. Sternite 7 and lateral parts of tergite 6 shiny. Acrostichals and dorsocentrals multiserial, not distinctly separated by bare stripe. Basal part of hypandrium with approximately eight setae arranged in an irregular transverse row, right ventral lobe of surstyli with very long seta (Figure 6a)…………………………………………………………………………. ***turcicus* sp. nov.**
–Fore tibia without conspicuously long dorsal preapical seta. Remaining characteristics different..……………………………………………………………….…….44.(3) Dorsocentrals uniserial, acrostichals biserial (sometimes with zero to six smaller setae irregularly arranged between rows). Labrum very short (usually less than 0.20 mm long), shorter than palpus. Palpus slightly swollen apically, with strong setae. (Additional characters: fore femur with a row of posterodorsal setae at least as long as femur depth; basal process of phallus short or absent; left ventral lobe of surstylus long and narrow—Figure 11b and [1] Figure 16)..………….……………………...….…5
–Dorsocentrals irregularly biserial, acrostichals quadriserial (setae in inner rows as subequally long as those in outer rows). Labrum longer than palpus. Palpus only slightly widened apically, covered with fine setae..………………………….65.(4) Antennal stylus shorter than postpedicel. Phallus thick with basal process short, nearly square-shaped [1], Figure 19. Larger species (wing usually more than 2.0 mm). (Additional characteristic: hypoproct lobe longer than cercus.).………………………………………………….…………..…***holosericeus* (Meigen)**
–Antennal stylus longer than postpedicel. Phallus thin without basal process (Figure 11f). Smaller species (wing usually less than 2.0 mm). (Additional characteristics: hypoproct lobe shorter than cercus.).……………………...…***strobli* Chvála**6.(5) Halter yellow. Hypandrium with 18 setae (see [17], Figure 3).….***zimini* Shamshev**
–Halter black. Hypandrium usually with 4–10 setae.……………………………….77.(6) Both mid and hind femora with conspicuously strong and long posteroventrals throughout whole length (7–10 setae on mid femur up to 1.5X longer than femur depth). (Additional character: stylus two thirds as long as postpedicel; abdomen almost velvety brown; dorsal lobes of surstyli twice broader in basal half and narrowed in apical half—Figure 4a,b).………………………………………….***nevadensis* sp. nov.**
–At most mid femur on distal third with conspicuously strong and long posteroventrals.……………………………………………………………………….88.(7) Mesoscutum in lateral and anterior views light, almost bluish grey. Mid femur with a row of posteroventral setae shorter than femur depth. (Additional characteristics: right ventral postgonite lobe nearly triangular, dorsal lobes of surstyli doubled on both sides—Figure 10a–c.).…………………………………..…..…***rostellatus* Loew**
–Mesoscutum in anterior view deep velvety black with two lighter narrow stripes along rows of dorsocentrals; in lateral view, deep velvety black. Mid femur without posteroventral setae on proximal two-thirds contrasting with several strong posteroventrals longer than femur depth on distal third to fourth. ……………………………………………………………..…………***baechlii* sp. nov.**9.(8) Halter yellow.………….…………………………………….…..……………………10
–Halter brown to black.……………………………………..……………….………1110.(9) Labrum half as long as head height. Fore femur with posteroventrals as long as femur depth. (Additional characteristics: acrostichals two to three serial; all abdominal sternites with long setae; stylus approximately half as long as postpedicel; hypandrium with two pairs of setae—see [16], Figures 5 and 7.).……………………………………………………….……………***gissaricus* Shamshev**
–Labrum nearly as long as head height. Fore femur with posteroventrals twice as long as femur depth. (Additional characteristics: stylus approximately two thirds as long as postpedicel; hypandrium with three pairs of setae—Figure 3d,e.) ……………………………………...…………………………..***chvalai* sp. nov.**11.(9) Acrostichals biserial, dorsocentrals uniserial. Stylus at least as long as postpedicel. ……………………………………………..…….…………………………***pallipes* sp. nov.**
–Acrostichals three to four serial, dorsocentrals multiserial. Stylus shorter than postpedicel.……………………………….………………………………………….1212.(11) Sternite 1 with one to three pairs of long median setae, sternites 2–3 each with at least three pairs of long setae, and mesoscutum in anterior view grey with black stripes below rows of acrostichals and dorsocentrals..……***anomalus* (Meigen)** (part.)
–Not as above..………………………………..……….…….……………..………….1313.(11) Right ventral postgonite lobe A-shaped (Figure 9d); dorsoapical lobes of hypandrium rounded (Figure 9b,e).……………………………...….***intermedius* Collin**
–Right ventral postgonite lobe S-shaped; dorsoapical lobes of hypandrium angulate…………………………………………………………………………………….1414.(13) Basal process of phallus bowed (Figure 8f)……….……………***crassipes* Macquart**
–Basal process of phallus nearly straight…..……….……………………..…..…….1515.(14) Dorsal processes of surstyli much shorter than epandrial lamellae depth and without conspicuous dorsal wart (Figure 7b); dorsoapical lobes of hypandrium with, at most, very short inconspicuous seta (Figure 7d,e); phallus process thin (Figure 7f)……………………………………………………...………..***anomalus* (Meigen)** (part.)
–Dorsal processes of surstyli as subequally long as epandrial lamellae depth and with conspicuous dorsal wart (Figure 12a,c; dorsoapical lobes of hypandrium with a long seta on both sides (Figure 12b,e); phallus process relatively thick (Figure 12f).…………………………………..……………..……….….***Microphor* sp.**



**Notes on some other species and faunistic records**


(“Faunistic records” paragraphs represent new country records, “Material examined” represent selected materials from CULSP collections.)


**
*Microphor anomalus*
**
**(Meigen, 1824)**


Figure 7.

**Faunistic records**: **Turkey**: Akyaka, river bank + salty meadow, 37°03′16″ N, 28°19′57″ E, Barták, Kubík, 16.-27.v.2011, 2 ♂; Toparlar, lowland forest, 36°58′39″ N, 28°39′30″ E, sweeping, 5.-7.v.2013, 7 ♂; same locality, 28.-30.iv.2016, 12 ♂; Akyaka, pasture, 37°03′19″ N, 28°20′07″ E, 28.iv.-8.v.2013, 6 m, 6 ♂; same locality, 27.iv.2016, 3 ♂; Akyaka, forest, 37°03′16″ N, 28°19′35″ E, 30.iv.-9.v. 2013, 30 m, 1 ♂; same locality, 26.iv.2014, 2 ♂—CULSP.

**Remarks:** Palaearctic, known from most European states (Austria, Bulgaria, Croatia, Czech Republic, Denmark, England, France, Germany, Hungary, Italy, Lithuania, Macedonia, Netherlands, Poland, Romania, Russia, Slovakia, Slovenia, Sweden, Switzerland, and former Yugoslavia). (In Fauna Europea, east Palaearctic is also included; however, we have not found any relevant record.) The first records are from Turkey.

**Neotype designation**: *M. anomalus* was described (as *Platypeza anomala*) according to a single specimen. Neotype (male), Czech Republic, Duchcov—2 km E, willow shrubs, 50.37 N, 12.43 E [=50°37′ N, 12°43′ E], 240 m, Barták, 16.vi.1993, deposited in CULSP. For the purpose of continuity, we selected the specimen exactly agreeing with the description and key by Chvála [1], who first revised the genus. We selected the specimen fulfilling all “typical” characters of this species mentioned by [1,18] and later supplemented by Martin Drake (pers. comm.), who observed materials deposited in Collin′s collection in Oxford University Museum of Natural History. These characters are specified in the key above (mesoscutum in anterior view pale grey with dark stripes along acrostichals and dorsocentrals, first abdominal sternite with long setae in middle sternites 2–4 with eight pairs of long straggling setae, mid femur with eight posteroventrals). The name-bearing holotype was lost (Daugeron, pers. comm. 2021; Igor Shamshev, pers. comm. 2021; Peter Sehnal, pers. comm. 2022) and neither Collin [18], nor Chvála [1,4] have mentioned anything about it. Collin [18] says that he had seen Meigen’s specimen of *M. holosericeus* at Paris, but does not mention having seen “anomalus”, so perhaps the type has been lost for a long time. The designation of a neotype is necessary due to the great variability of the species that contains possibly several siblings, and in order to stabilise the usage of the name consistent with previous authors (see discussion).

Description of genitalia (Figure 7): dorsal lobe of both (right and left) surstyli short and narrow, slightly bowed, (0.08–010 mm long; Figure 7a,b; however, some specimens were studied with somewhat longer lobes with very small and scarcely visible dorsal wart under Mag. 120X). Hypoproct moderately sclerotized, shorter than cerci. Middle lobe of right surstylus sharply triangle shaped, left—oblique rectangle to almost triangle-shaped, both very short and visible only in dorsal views (not illustrated here). Both apicodorsal lobes of hypandrium with short hair, right apicoventral lobe of hypandrium long, narrow and irregularly bow-shaped (folded at approximately middle, Figure 7e). Phallus (Figure 7f) simply bowed, straight at apex (rarely slightly narrowed), and phallic process narrow tipped and nearly straight (slightly differing in length and width between specimens). Right ventral postgonite lobe S-shaped (Figure 7c). Genitalia are strikingly similar to Nearctic *M. obscurus* ([6], Figure 69–74) and further studies are necessary to elucidate their possible conspecificity.


**
*Microphor crassipes*
**
**Macquart, 1827**


Figure 8.

**Material examined**: Czech Republic: Vráž nr. Písek, damp meadow, 49°23′ N, 14°8′ E, 400 m, Barták, 1.vi.1992, 1 ♂; same locality, 31.v.1993, 2 ♂; Kunice, Vrchy, edge of mixed wood, 49°56′ N, 14°42′ E, 380 m, Barták, 5.vi.1988, 1 ♂; Kozlov—2 km E, meadow nr. picetum, 49°24′ N, 15°41′ E, 340 m, Barták, 27.v.1986, 1 ♂—CULSP.

**Remarks**: Figures 44–47 by Chvála [1], identical with Figures 628–631 in Chvála [4], do not correspond with our specimens (the basal phallic process is straight), probably because his figures are not adequately detailed (e.g., the surstyli lobes are usually not illustrated) and these figures probably belong to a wrongly identified specimen of *M. anomalus*. The genitalia are therefore described here for the first time (Figure 8): dorsal lobe of both surstyli (Figure 8a,b) rather long and thick (0.12 mm long), slightly bowed, with dorsal hairy wart. Hypoproct medium sclerotized, shorter than cerci. Internal lobe of right surstylus (not illustrated here) triangle-shaped, left—oblique rectangle, both very short. Right apicodorsal lobe of hypandrium with small process ending with short hair (Figure 8e), left apicoventral lobe right-angled, right apicoventral lobe of hypandrium long, narrow, and irregularly bow-shaped (Figure 8e). Phallus (Figure 8f) simply bowed (with inconspicuous angle submedially); at apex, narrowed and straight, phallic process strongly bowed basally and almost parallel-sided. Right ventral postgonite lobe narrowly S-shaped (Figure 8c).


**
*Microphor holosericeus*
**
**(Meigen, 1804)**


**Faunistic records: Turkey**: Turkey:13 km NE of Muğla, pine wood + pasture, 37°15′ N, 28°30′ E, 1100–1300 m, Barták, Kubík, 2.-3.v.2016, 3 ♂, 6 ♀; **Portugal**: Trás-os-Montes, Esinhosela, pasture, 790 m, 41°53′8″ N, 6°49′37″ W, A. Gonçalves, 16.v.2015, 1 ♂, 4 ♀; Trás-os-Montes, Gondesende, nr. river, 640 m, 41°50′47″ N, 6°52′46″ W, A. Gonçalves, R. Andrade, 10.v.2015, 3 ♂, 1 ♀; Trás-os-Montes, Esinhosela, pasture, 790 m, 41°53′8″ N, 6°49′37″ W, A. Gonçalves, 16.v.2015, 1 ♂—CULSP.

**Remarks:** A broadly distributed species, their records are from Palaearctic: Algeria, Austria, Bulgaria, Danmark, England, Finland, France, Germany, Greece, Hungary, Ireland, Italy, Lithuania, Macedonia, Mongolia, Netherlands, Norway, Poland, Romania, Russia, Slovakia, Spain, Sweden, Switzerland, and Ukraine. (In Fauna Europea, the near east is also included; however, we have not found relevant record.) The first records are from Turkey. The genitalia of this species were illustrated in detail by several authors [9,19]. Some variation in structures is mentioned in the discussion.


**
*Microphor intermedius*
**
**Collin, 1961**


Figure 9.

**Material examined**: Czech Republic: Třinec, Sosna, edge of wood, YPWT, 49°46′ N, 18°44′ E, 350 m, 24-27.vii.1996, Barták 1 ♂; Strachotín, 48°54′N, 16.39 E, 12.v.1982, Barták, 1 ♂; Bystřice nr. Třinec, shores of Olše river, SW, 49°38′8″ N, 18°42′26″ E, 27.-28.v.2005, Barták, 2 ♂; Bílina, Holibka, nr. pond, YPWT, 420 m, 50°31′20″ N, 13°49′40″ E, Kubík, 30.v.-1.vi.1997, 1 ♂; Bulgaria: Gorno Sahrane, pasture nr. river, 42.633° N, 25.219° E plus/minus 1 km, Barták, Kubík, 30.iv.-9.v.2018, 1 ♂; Croatia: Gorni Muć, abandoned garden, 500 m, MT, 43°41′27″ N, 16°29′44″ E, 500 m, 27.v.-22.vi.2013, B. Kokan, 1 ♂; same locality, 27.iv.-10.v.2014, B. Kokan, 1 ♂; 3 km N of Molunat, meadow nr. wood, 150 m, 42°28′52″ N, 18°25′57″ E, Barták, 21.v.2007, 1 ♂—CULSP.

**Remarks**: This species was inadequately described [18] and our conception of its identity is based on an examination of Collin′s types by Martin Drake,on Chvála′s [1] Figures 39–43, and on the fact that several specimens from CULSP materials (but not all) have a conspicuously narrower hindleg than the remaining species. This characteristic is therefore considered as variable. The genitalia are as in Figure 9: dorsal lobe of both surstyli narrow, rod-like ca. 7×x longer than wide (0.07 mm long), with an additional smaller lobe (0.02 mm long). Hypoproct medium sclerotized, shorter than cerci. Internal lobe of left surstylus triangle shaped, right—oblique rectangle, both very short. Both apicodorsal lobes of hypandrium broadly rounded (with practically invisible short seta), right apicoventral lobe of hypandrium long, narrow, and simply bow-shaped. Phallus simply bowed; at apex, narrowed and slightly upcurved, phallic process straight, long, and narrow. Right ventral postgonite lobe inverted “1”-shaped.


**
*Microphor rostellatus*
**
**Loew, 1864**


Figure 10.

**Material examined**: France, 9 km S of Sété, seashore dunes, PT [=white and yellow water pan traps], 43°21′9″ N, 3°45′48″ N, M. Barták, 21-23.v.2009, 9♂, 9 ♀.

**Remarks**: The illustrations of Chvála [1], Figures 22–27, are rather inadequate. For this reason, the first detailed illustration and description is given here. Both left and right epandrial lobes (Figure 10a,b) are very similar: the dorsal surstylus of both doubled, inner (probably median) lobe “V”-shaped is forked, and both tips end in a spine (altogether, two spines are apparent in the lateral view). Postgonites with apodemes slightly leaf-shaped widened, dorsal lobes strongly asymmetrical, right one broad and left one very narrow (Figure 10c). Hypandrium with left ventral lobe not prominent (Figure 10d), right one slightly prominent (Figure 10e); dorsal lobes connected with postgonites with narrow rod-like structures. There are three pairs of hypandrial setae.


**
*Microphor strobli*
**
**Chvála, 1986**


Figure 11.

**Faunistic records: Bulgaria**: 5 km W of Smolyan, glade, sw [=sweeping vegetation], 1260 m, 41.569 N, 24.632 E, Barták, Kubík, 14.-17.vi.2019, 1 ♂; same locality, 14.-26.vi.2019, MT, 1 ♂; same locality, 23.vi.2018, 1 ♂; same locality, 25.-26.vi.2016, 19 ♂, 7 ♀; Pamporovo, 1300–1600 m, sw + pt [= sweeping vegetation and white or yellow pan water traps], 41.639 N, 24.697 E, Barták, Kubík 14–18.vi.2018, 1 ♂, 1 ♀; 25 KM SSW of Plovdiv, 1590 m, meadow, 41°56′5″ N, 24°40′45″ E, Barták, Kubík, 20.vi.2016, 2 ♂—CULSP.

**Remarks:** A little known species, from Palaearctic: Austria, Czech Republic, Germany, Central European Russia, Slovakia, and England. The first records are from Bulgaria. The genitalia are herewith firstly illustrated in detail (Figure 11).

Both epandrial lamellae have a short dorsal sursylus lobe, and the left lamella has a prominent ventral lobe and strongly developed hook-shaped middle lobe (Figure 11b). The hypandrium has three pairs of setae, and both ventroapical lobes are broadly rounded (Figure 11d,e). The phallus is narrow, without a ventral process (Figure 11f). The left ventral postgonite lobe is unusually broad (Figure 11c).


***Microphor* sp.**


Figure 12.

**Material examined**: **Bulgaria**: Rhodopes Yundola, 1300 m, pasture, 42°3′47″ N, 23°51′17″ E, Barták, Kubík, 30.vi.2016 1 ♂—CULSP.

**Remarks**: An unnamed species from the *M. anomalus* complex. We hesitate to describe this specimen as a new species because of the slight variability observed in details of the genitalia of “typical” *M. anomalus* (e.g., in the length and exact shape of dorsal surstyli lobes, phallus process, and tip of phallus). The unnamed species is very similar to the remaining species of the *M. anomalus* complex; only the posteroventral setae on the hind femur are conspicuously long (longer than the femur depth) and sternite 1 bears one lateral seta. The genitalia are as in Figure 12: the dorsal lobe of both surstyli is very long and broad, with dorsal warts (Figure 12a,c). The hypoproct is poorly sclerotized and half as long as the cerci. Both middle surstyli lobes are sharply triangle-shaped and very short. Both apicodorsal lobes of hypandrium angulate and have a long seta (Figure 12b,e), and the right apicoventral lobe of the hypandrium is short and irregularly curled. The phallus is simply bowed; at the apex, it is straight, and the phallic process is very broad with a narrower tip that is nearly straight (Figure 12f). The right ventral postgonite lobe is S-shaped (Figure 12d).

## 4. Discussion

The genus *Microphor* contains small to very small true flies (Diptera) well known for the kleptoparasitic behavior of females, often observed “stealing” tiny prey from spider webs. *Microphor* remains a broadly defined group whose phylogenetic background is not quite clear. The recently published revision of North American species [6] provides the first provisional evidence for the monophyly of *Microphor*, especially with respect to *Schistostoma*.

Changes in the taxonomic classification of the Microphorinae genera that have taken place in recent times indicate the confusion that this group arouses. Therefore, many authors accepted the solution proposed by Sinclair and Cumming [5]. However, there is not much difference between accepting a separate family and including it in Dolichopodidae *sensu lato*.

There are problems with the delimitation of some species of *Microphor*. Variability not only in external characters, but also in the male genitalia, had already been noticed by Ulrich [8], who described in detail the postabdomen of *M. holosericeus* and found two forms, which he did not formally describe as separate species. Experts in Belgium are currently studying this issue (Patrick Grootaert, pers. comm. 2021). Several specimens of this species from the CULSP collection have four light bluish-grey longitudinal stripes on the mesoscutum. However, the similarity of the unique male genitalia of this species exactly agrees with the more common form (with a dark black mesoscutum without any stripes), and therefore we consider them conspecific.

We found some small variation in the shape of the male genitalia of *M. anomalus*. By dissecting many individuals, we noticed small differences in the shape and length of the dorsal surstylar lobes in particular. Interestingly, this is the same characteristic differentiating “forms” in *M. holosericeus*. We suppose that this structure may be subject to variability (a similar situation is observed in the shape of the left cercus in many species of *Platypalpus*—personal observation). The variability between individuals of *M. anomalus* is not only seen in the male genitalia, but also among external characteristics, most notably in the coloration of the mesoscutum. The mesoscutum of most specimens of *M. anomalus* appears in the anterior view as light grey with distinct dark black stripes below the acrostichals and dorsocentrals (or only below dorsocentrals), turning to black in dorsal and posterior views (sometimes with two paler stripes between the rows of setae), but many specimens (more frequently from southern countries) have the mesoscutum as black from all points of view. Most specimens (but not all) have one to three pairs of strong setae in the middle of sternite 1 and relatively short posteroventral setae on the mid femur that are only slightly longer than the femur depth. Usually, three or more pairs of long setae on abdominal sternites 2–4 are considered “typical” for *M. anomalus* (e.g., [1]), but we studied several specimens with shorter and less numerous setae on the abdominal sternites. The remaining external characters (number and length of acrostichals and dorsocentrals, width of wing stigma, and length of antennal stylus) are variable.

From the above arguments, we consider that specimens of the *M. anomalus* complex may be identified only by a detailed study of the male genitalia. For this reason, many countries’ records should be re-examined. However, we cannot exclude that, in fact, this complex represents a broader complex of sibling species.

It seems that most West Palaearctic species can be determined reliably by external characters, except for the *M. anomalus* complex. We identified many individuals of this complex using male genitalia and then evaluated several external characters as accurately as possible (e.g., the presence of ventral setae on sternite 1, length of acrostichal setae, ratio of length of stylus/ postpedicel, length and number of posteroventral setae on mid femur, length and number of acrostichals and dorsocentrals, etc.), and all values showed variability and overlapped. For these reasons, it would be necessary to re-examine the correctness of the identification of the species, especially regarding the *M. anomalus* complex, and to adjust records of their distribution.

## Figures and Tables

**Figure 1 insects-13-00700-f001:**
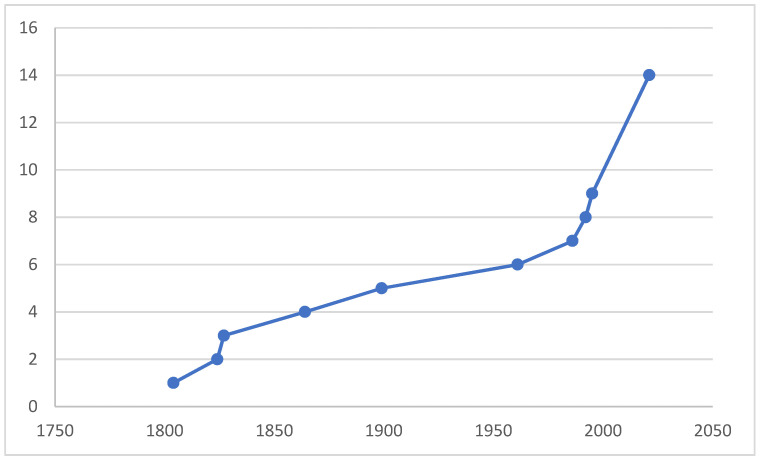
A cumulative curve of number of valid species of *Microphor* described from West Palaearctic, incl. two species from Central Asia.

**Figure 2 insects-13-00700-f002:**
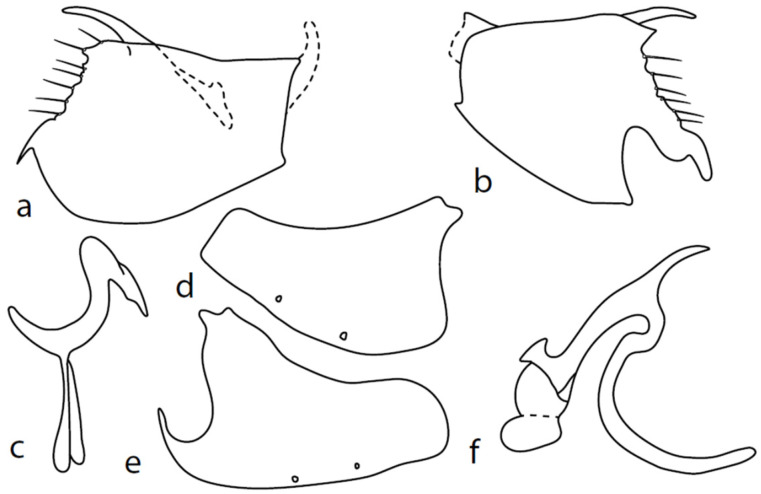
*M. baechlii* sp. nov.: (**a**) = right epandrial lamella, (**b**) = left epandrial lamella, (**c**) = postgonites from left, (**d**) = hypandrium from left, (**e**) = hypandrium from right, (**f**) = phallus from left. Dots on (**d**,**e**) are insertion points of large macrochaetae.

**Figure 3 insects-13-00700-f003:**
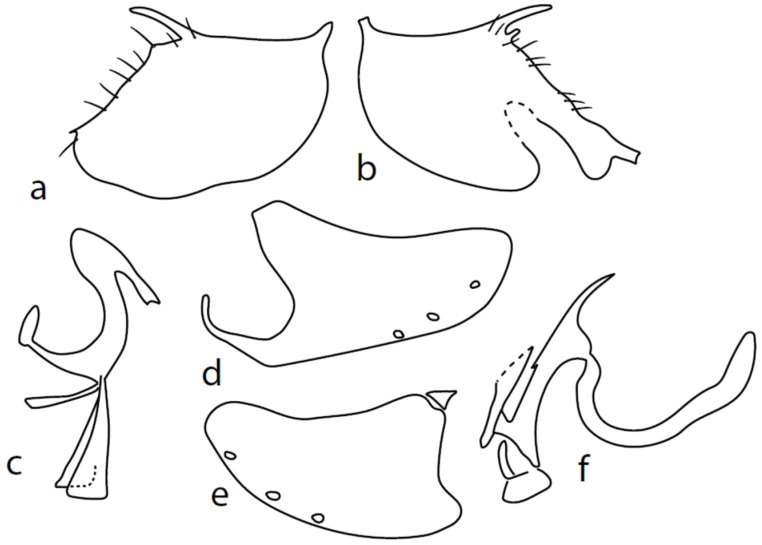
*M. chvalai* sp. nov.: (**a**) = right epandrial lamella, (**b**) = left epandrial lamella, (**c**) = postgonites from left (left dorsal process without connecting structure, which remained attached to left dorsoapical hypandrial lobe), (**d**) = hypandrium from right, (**e**) = hypandrium from left, (**f**) = phallus from left.

**Figure 4 insects-13-00700-f004:**
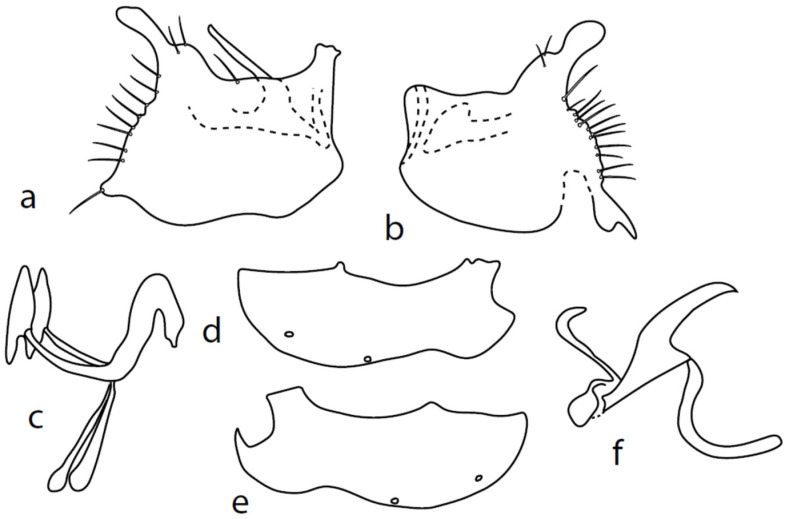
*M. nevadensis* sp. nov., genitalia: (**a**) = right epandrial lamella, (**b**) = left epandrial lamella, (**c**) = postgonites, (**d**) = hypandrium from left, (**e**) = hypandrium from right, (**f**) = phallus from left.

**Figure 5 insects-13-00700-f005:**
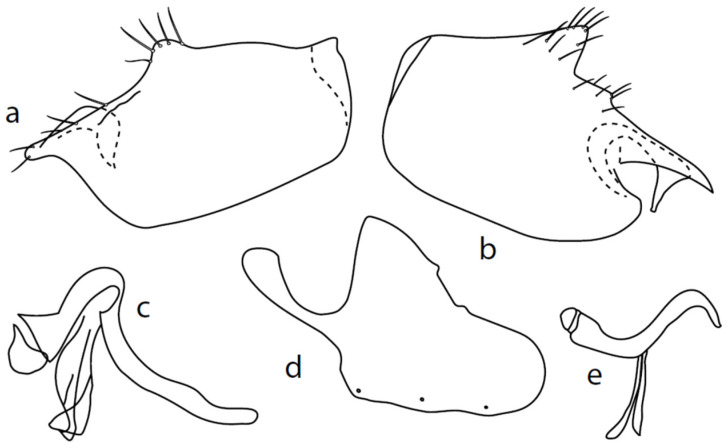
*M. pallipes* sp. nov.: (**a**) = right epandrial lamella, (**b**) = left epandrial lamella, (**c**) = phallus from left, (**d**) = hypandrium from right, (**e**) = postgonites from left.

**Figure 6 insects-13-00700-f006:**
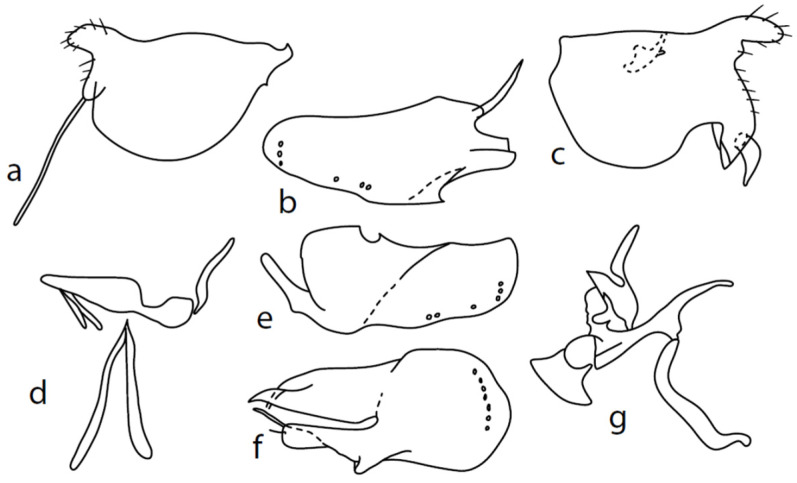
*M. turcicus* sp. nov.: (**a**): right epandrial lamella, (**b**) = hypandrium from left, (**c**) = left epandrial lamella, (**d**) = postgonites, (**e**) = hypandrium from right, (**f**) = hypandrium ventral view, (**g**) = phallus from left.

**Figure 7 insects-13-00700-f007:**
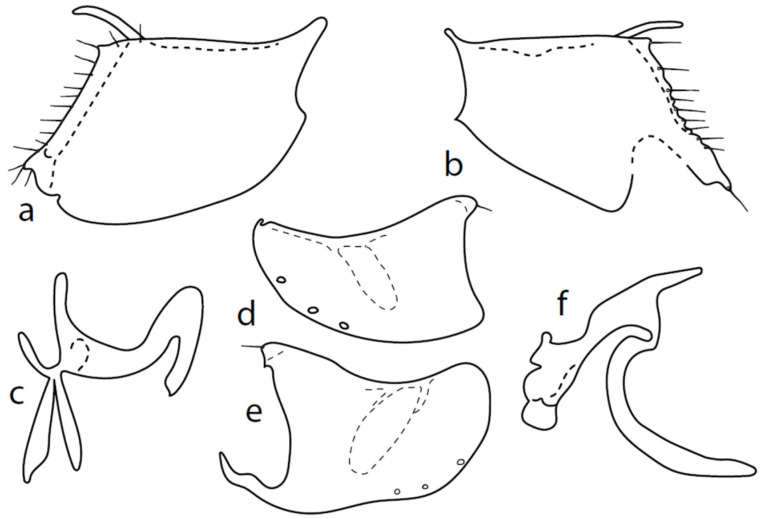
*M. anomalus*: (**a**) = right epandrial lamella, (**b**) = left epandrial lamella, (**c**) = postgonite from left, (**d**) = hypandrium from left, (**e**) = hypandrium from right, (**f**) = phallus from left.

**Figure 8 insects-13-00700-f008:**
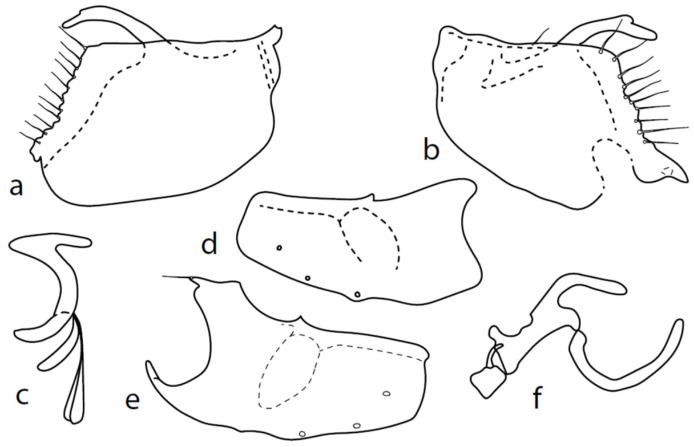
*M. crassipes*: (**a**) = right epandrial lamella, (**b**) = left epandrial lamella, (**c**) = postgonite from left, (**d**) = hypandrium from left, (**e**) = hypandrium from right, (**f**) = phallus from left.

**Figure 9 insects-13-00700-f009:**
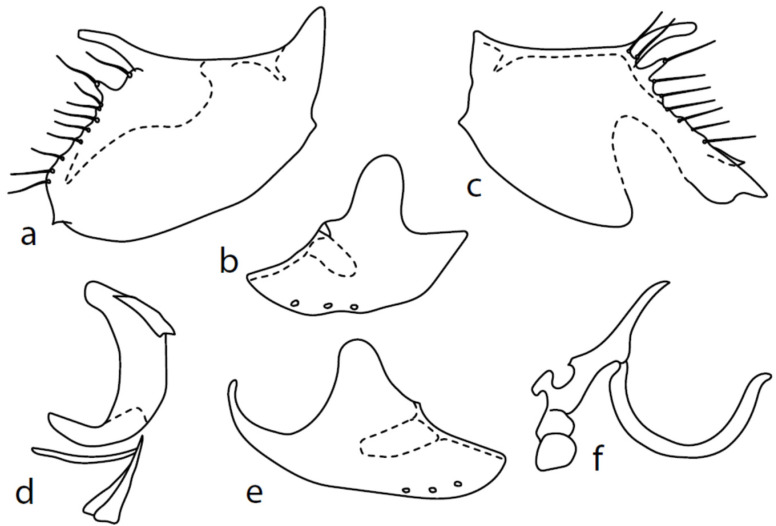
*M. intermedius*: (**a**) = right epandrial lamella, (**b**) = hypandrium from left, (**c**) = left epandrial lamella, (**d**) = postgonite from left, (**e**) = hypandrium from right, (**f**) = phallus from left.

**Figure 10 insects-13-00700-f010:**
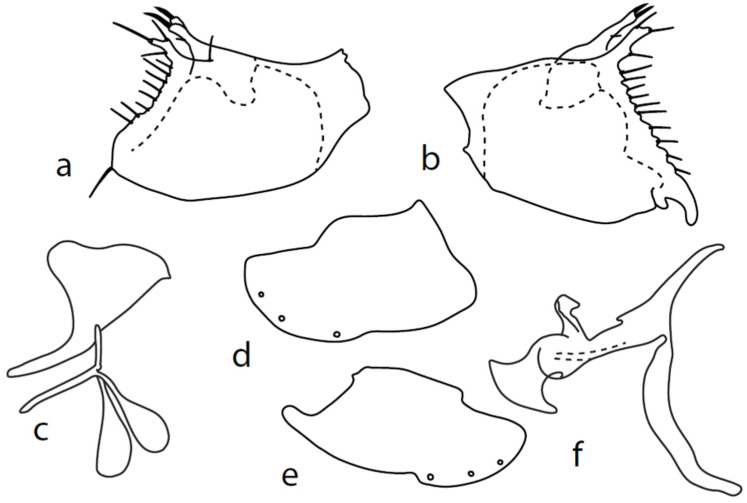
*M. rostellatus*: (**a**) = right epandrial lamella, (**b**) = left epandrial lamella, (**c**) = postgonite from left, (**d**) = hypandrium from left, (**e**) = hypandrium from right, (**f**) = phallus from left.

**Figure 11 insects-13-00700-f011:**
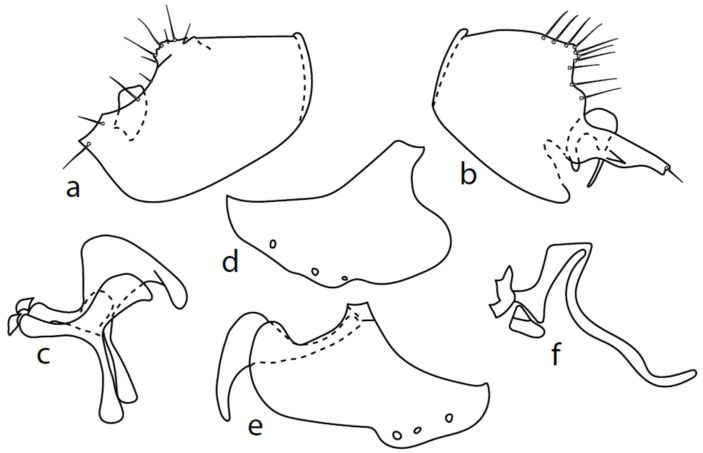
*M. strobli*: (**a**) = right epandrial lamella, (**b**) = left epandrial lamella (with enlarged middle postgonite lobe), (**c**) = postgonite from left (left ventral postgonite lobe strongly developed), (**d**) = hypandrium from left, (**e**) = hypandrium from right (right dorsal postgonite lobe attached to right apicodorsal lobe of hypandrium), (**f**) = phallus from left.

**Figure 12 insects-13-00700-f012:**
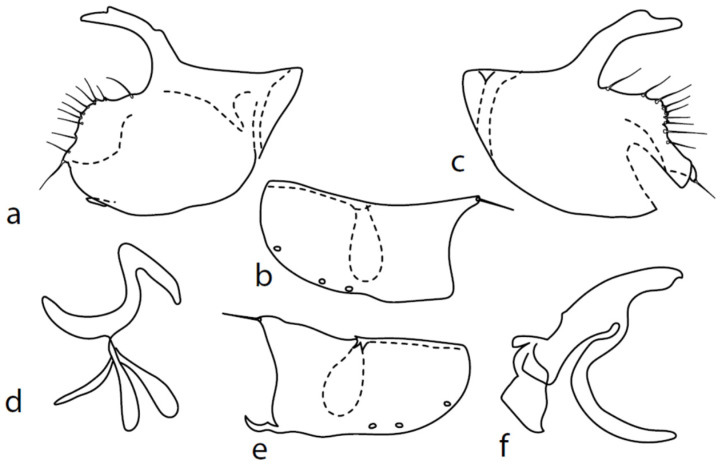
*Microphor* sp.: (**a**) = right epandrial lamella, (**b**) = hypandrium from left, (**c**) = left epandrial lamella, (**d**) = postgonite from left, (**e**) = hypandrium from right, (**f**) = phallus from left.

## Data Availability

Data are contained within the article. The data presented in this study are available in this article.

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
