# Peer review of "Unexpected Previously Unknown Diversity of the Genus Microphor Macquart (Diptera: Dolichopodidae: Microphorinae) in the West Palaearctic"

_insects, 2022, doi:10.3390/insects13080700_

Round 1
Reviewer 1 Report
Comments on a paper submitted to the journal Insects by M. Barták and Š. Kubik:
Unexpected previously unknown diversity of the genus Microphor … in the West Palaearctic.
General remark: I am not a native English speaker. Nevertheless, I have some practice in English speaking and writing and therefore I allow me to remark, that there are – in my opinion – some constructions used in the text – which are at least not well sounding. If I am wrong with my opinion, I bag your pardon, but I would propose, that at least the introducting chapters should be checked by a native speaker.
As to the title: I am not sure, if the term “West Palaearctic” is correct, as in most papers on this region of the Palaearctic Realm authors write “Western Palaearctic”. (In Websters Dictionary as “Western” is called a region lying towards the West, hence it should sound “Western Palaearctic”, I guess).
Yes, indeed there are some astonishing findings and the title gives a suggestion, that the reader may await descriptions of several new species. It is therefore not titled as a revision of the Western Palaearctic species. Please allow later on to loose some lines to this problem. (under M. anomala).
Simple Summary: Frankly spoken I wonder about some of the sentences used in the “Simple Summary” (and these are found in the “Introduction” as well [lines 59 to 62]): The declaration, that this paper (with its undoubtedly valuable taxonomic contents) is written “in the light, that the unprecedent reduction in biodiversity and the further mass extinction ….” etc. is in my point of view at least misleading. [Indeed this conclusion or attitude to do description work is needing a more philosophical discussion and I guess, that this taxonomic paper might not be the right place, to discuss this matter with the necessary diversity of arguments.] Why taxonomists describe new species? If this is a question in need to discuss, it should be done in a separate chapter “Discussion”. This point (Discussion) is missing at all! Later on, I will give some arguments, why such a chapter should urgently be added.
I come back to the question of “Why to describe species?”. It is undoubtedly the fact, that scientists are generally driven in their work by a force which may be summarised as “practicing science to gain knowledge about the world”. Under the stipulation given in this paper, it might for instance be rather “senseless” to describe e. g. the diversity of fossil diptera in amber. I mean, that a taxonomical work takes its sense from itself. There is no need to give a special declaration! On the other hand (and this is an “implication” given by the declaration in question) it is not sure, if (some) Microphor species are threatened by extinction. Only one instance: Falk & Crossley (2005) excluded Microphor crassipes from the former British “Red data Book” because it is found in more than 14 British countries actually; in the former edition of that list it was called “notable” (Falk, S.J. & Crossley, R. 2005. A review of the scarce and threatened flies of Great Britain. Part 3: Empidoidea. Species Status No. 3, JNCC, Peterborough, 134 pp.). This “new” understanding is – on the other hand – indeed an argument, that we should not stop to do faunistical research (once the species are described!) => which is also self-explanotary. May be, that one or another species described in this paper is “endangered” due to the special type of biotope, it is inhabing, but there is no evidence for this. If this is a point, which is worth to be discussed from the point of authors’ opinion it should be added in the discussion chapter. A paper which might serve to be cited accordingly to this theme is https://onlinelibrary.wiley.com/doi/epdf/10.1111/brv.12816
(Cowie et al. 2022, free access at Wileys)
To “1. Introduction”. In line 34 is written “The Microphoridae have been elevated to subfamily rank by Chvála …” (1983). This is wrong. Chvála (1983) raised the Microphorinae to family level (Microphoridae) (p. 64). => To say it clearly: [Higher level] systematics of this “group” of flies is in my point of view one of the most complicated in the Empidoidea (or even Diptera in general). It is in the same category as e.g. the Platypezidae-Opetidae systematics is. I do not want to discuss the question here, if the “species” is the only “true” systematic unit and so on. There is for sure a need to enlight the monophyletical groups within the organisms and therefore the thorough studies of Hennig, Ulrich, Chvála, Sinclair and Cumming, Shamshev and Cumming, Brooks and Cumming (all without citations here in this review) led to the broadly accepted finding, that Microphor is a “part” of the Dolichopodidae (by the way, the impulse to deal more thoroughly with this matter was Ulrich in his papers on morphology of Mirophor!). Without any doubt, all species treated in the current paper are belonging to this genus.
In line 40 the term “modern” is used to characterise extant (Recent) species. I would recommend to avoid this term here, because “modern” is used in weighting and discussing characters in a phylogenetical sense.
Line 50: Here the authors state, that “Characters of genitalia are the most important for distinguishing of species.” I strongly recommend to add a chapter “Discussion” in which this statement should be reflected on. Other workers on the genus (especially Chvála) focused also to the shape of the antenna (relations between postpedicellus and stylus, shape of postpedicellus etc.), length and number of setae on thorax, shape of thorax (strongly humpbacked or not) etc. pp. I wished to have an explanation, which characters are senseful in the light of alpha-taxonomy, and which are not. It is clear, that in nearly all Insect groups one may use the genitalia to determine a species, but if there are some other accessory characters (which might be unique in comparison with others) I would use those first (e.g. yellow legs of M. pallipes spec. nov.), to determine the species. [I am aware, that this is also a point of discussion if keys have to be “phylogenetical” or “practicable”].
The caption to the figure should sound (proposal): Cumulative number of valid species of ?taxonomical unit? From the Western Palaearctic including two species from Central Asia.
The headline in the figure itself should be omitted.
To “2. Materials and Methods”: It is clear that such a special paper has a target group of readers, which have a certain knowledge of the anatomy of the group in question. Nevertheless I find it important, to have an introducing figure (as citation from other authors?) which shows the (complicated) spatial arrangement of anatomical parts of the male genitalia. [I could offer a high resolution photograph as well, see appendage] (=> if there arise costs one may use my “concessionary” which I have at MDPI which I gained for earlier revisions)
[Remark => do not want to be co-author!] In any case: A figure (may be a line drawing citation of that one given by Hennig (should be added here or in the first chapter of “Results” under “Taxonomy” [By the way: later on there is a lack of a drawing of M. holosericeus genitalia, which I strongly recommend to have. The inclusion of the photograph delivered here would avoid high efforts in this phase of the submission of the paper to make drawings of the parts of this (M. holosericeus) hypopygium as well.
As to naming of the different parts, I wrote an e-mail to Bradley Sinclair (as one of the most skilled workers in this field) which sounds as follows:
“… please allow to contact you on behalf the following issue. I was asked to do a referees comment to a paper in which Bartak and Kubik are describing several new Microphor species. In their drawings the use the following parts to distinguish the species: right and left epandrial lamellae, views of the hypandrium from left and right, postgonite, phallus (I append one of these drawings as instance). In my opinion the paper is a thorough elaboration but there is a lack of an introducing "figure" which might give an imagination of the complicated anatomy of the several parts in a three-dimensional view. On the other hand, a year ago, I found during my faunistical investigations in Saxony-Anhalt an obviously sibling species of M. holosericeus (but not identical with the "form A or B", used by Ulrich 1988 in his studies on the hypopygium of M. holosericeus.). I hesitated to describe the species because I have heard, that other groups of workers were dealing with Microphor and I do not wanted to "intrude" their work, rather waiting, if they include my species as well. Nevertheless I took many photographs of the hypopygia etc. (macerated in concentrated lactic acid and then bleached with H2O2.) As in the paper of Bartak and Kubik such a figure is lacking, I wanted to offer this one for their paper. As I told you, I took the efforts before and therefore it would be fast to offer the figure. On the other hand, to name the parts is not easy at all!, and now my question goes to you. You surely know the paper of Hennig on the hypopygium of Lonchoptera (1976). There also a hypopygium of Microphor (holosericeus) is figured. My problem is, that I do not find in any of the "modern" papers (even not in Brooks and Cumming 2022) a clear name of the clasper-like structures, which I see very prominent in my photographs. Hennig says "telomers" to name them. This term I found together with "claspers, distimere, dististyle, stylus" set equal with gonostylus in Manual of Afrotropical diptera (Vol 1). On the other hand, I see here in this publication but also in other papers by your authorship, that these are not existing in the eremoneura. In Ulrichs elaboration (1988) one may only find the term "Drüsenkegel" which is in my opinion completely misleading as this structure is heavenly sclerotised and forms a pairy structure which may fix the females parts during copulation (clasping it) (compare the second figure of the obviousely undescribed species).
Brad Sinclair has forwarded my question to Scott Brooks => name of the parts in question: right and left dorsal lobes of the postgonites
To “3. Results”: In general: All descriptions should be checked, if there is information doubled with the diagnosis given for the genus. For instance the statement “Eyes without ommatrichia” should not be repeated in the species description. This is true for the colouration of head, thorax or abdomen (black?). The descriptions are somewhat too detailed but this might be a personal opinion. I do not really know, if the measurements of setae or the length of labrum given in mm (instead as relation to other parts) is necessary. Especially the mouthparts may be sunken within the head in dried specimens etc.
I also strongly recommend to change the appearance of measurement of the whole insect and the wing to the beginning of the description.
May be – if of further importance* -, the basic coloration might be given in a short table:
|
species/colour of certain part |
head |
thorax |
abdomen |
legs |
|
M. baechlii |
black |
black |
brownish black |
blackish brown |
|
M. chvalai |
brownish black |
black |
brownish black |
blackish brown |
|
M. nevadensis |
black |
black |
brown |
blackish brown |
|
M. pallipes |
brownish black |
blackish brown |
brown |
yellowish brown |
|
M. turcicus |
black |
black |
black |
black |
* I am in a doubt, if these fine gradations of coloration of head, thorax and abdomen are really “visible” and constant within the species.
Lines 125 to 187
I find this part too long and too much detailed. I am of the opinion, that no one has a profit, if the relations of parts between the species are given here. There is also no phylogenetical important meaning qualified by the authors of one or another structure and its “relations” (e. g. line 137 “… or twice thickened basally etc. pp.). I would propose to omit all these comparisons here. [I am aware that Brooks and Cumming have used a similar scheme but also in that paper I am in a doubt if this was necessary.]
Line 352
As to the fact, that there are Nearctic species of Microphor exist, I would recommend, not only to use “nevadensis” as species specific epithet (as far as the authors refer to the geographical origin – as is stated in line 354). In North America is also a “Sierra Nevada” which is also famous. To avoid “confusion” within the species of the genus (despite that there is no species named after the Californian Sierra Nevada) I would propose a name referring to the Spanish Sierra Nevada. (=> “hispanianevadensis” or somewhat “shortened” => “hispanevadiensis” or “hispaniensis”)
Microphor anomalus and M. holosericeus
According to the discussion within the characteristics of M. anomalus the authors are aware, that besides the clearly defined species they mention, this “species” may include sibling species as well. May be, that readers at the end having an idea, why authors write in line 726 that they choose a neotype which fulfills “all “typical” characters …” etc. but here no connection to the need of avoiding problems with “sibling” species is thoroughly stated. I wished to got mentioned that problem here in this paragraph also.
Ulrich (1988) mentioned two “forms” of M. holosericeus, which he has collected in his garden (and which were subsequently used for his anatomical work on the hypopygium of that species.) Ulrich hesitated to “describe” one or the other form (“A” and “B”) as certain species, most probably to avoid to search for the type of M. holosericeus and/or to open the full taxonomical cascade to designate a neotype etc. pp., as he (Ulrich) was mainly interested in the inner anatomy of the species to light up its phylogenetical position within the Empidoidea. In the “Schlussbemerkungen” Ulrich stated: “Die Befunde der vorliegenden Untersuchungen und der Vergleich mit den früher bearbeiteten Empidoidea belegen, daß das Hypopygium von Microphor holosericeus einen überdurchschnittlich verwickelten Aufbau hat.“ (p. 201) [„The findings of the present investigations and the comparison with the previously studied Empidoidea prove that the hypopygium of Microphor holosericeus has an above-average complicated structure.] In the paragraph to M. holosericeus the authors of the current paper do not discuss the situation in a thorough manner. Conclusion: I strongly recommend to add a (short) chapter “Discussion”, to state, that in general there might be several hidden Microphor species (in the vicinity of M. anomalus and holosericeus) – which are not as “good” characterised as the new ones described here. Translation of the paper of Hennig on Lonchoptera lutea: http://www.canacoll.org/Diptera/pdfs/Hennig_1976_The_hypopygium_of_Lonchoptera_lutea.pdf References: In this chapter some careless mistakes may be found, which I do not want to mention in detail. Genera are not given in italics (line 926, 953), hard hyphen not omitted (line 937). Should carefully be checked.

Author Response
All is in Word file

Reviewer 2 Report
Dear Authors,
This MS is of extremely high importance since it is dealing with a group of species on a very large geographical scale. It gives an insight into the unknown diversity of the genus Microphor and describes 5 new species, but also gives better descriptions of already known species. There are some small mistakes that need to be mended in the text and I wish for a short discussion to be written, as like this I feel like something is missing.
All in all the MS is very good.
Best wishes

Author Response
We changed everything in accordance with the reviews
Reviewer 3 Report
Greetings and Regards
The question that is raised is to clearly mention in the text what is the reason for the presence of these species? Does a certain plant attract this type of Microphor Macquart (Diptera: Dolichopodidae: Microphorinae)?
Has it been in a particular area? Or the climatic and regional conditions are the cause of this issue.
The text in figure 1 should be inserted in the appropriate font of the journal. There is also a need to update resources. And in terms of the fluency of the written text, it needs grammatical correction.
Author Response

(The authors gave the same response as above.)

Round 2
Reviewer 1 Report
Dear colleagues, I am sorry, to have forgotten during the first peer process to add an information on M. strobli. I append a paper, were the finding of this species is mentioned for Germany. May be that it is only sufficient to add "Germany" in the list of localities without mentioning this paper in detail.

Author Response

(The authors gave the same response as above.)

Reviewer 2 Report
The MS is very good now. I think it is ready for publicaion.
Author Response

(The authors gave the same response as above.)
